

# Flexible and Consistent Quantile Estimation for Intensity-Duration-Frequency Curves

Felix S. Fauer[1], Jana Ulrich[1], Oscar E. Jurado[1], and Henning W. Rust[1]

[1]Institute of Meteorology, Freie Universität Berlin, Carl-Heinrich-Becker-Weg 6-10, 12165 Berlin, Germany

**Correspondence:** Felix S. Fauer (felix.fauer@met.fu-berlin.de)

**Abstract.** Assessing the relationship between intensity, duration and frequency (IDF) of extreme precipitation is required for the design of water management systems. However, when modeling sub-daily precipitation extremes, there are commonly only short observation time series available. This problem can be overcome by applying the duration-dependent formulation of the generalized extreme value (GEV) distribution which fits an IDF model with a range of durations simultaneously. The

originally proposed duration-dependent GEV model exhibits a power-law like behaviour of the quantiles and takes care of a deviation from this scaling relation (curvature) for sub-hourly durations (Koutsoyiannis et al., 1998). We suggest that a more flexible model might be required to model a wide range of durations (1 min to 5 days). Therefore, we extend the model with two features: i) different slopes for different quantiles (multiscaling) and ii), newly introduced in this study, the deviation from the power-law for large durations (flattening). Based on the quantile skill score, we investigate the performance of the resulting

flexible model with respect to the benefit of the individual features (curvature, multiscaling, flattening) with simulated and empirical data. We provide detailed information on the duration and probability ranges for which specific features or a systematic combination of features leads to improvements for stations in a case study area in the Wupper catchment (Germany). Our results show that allowing curvature or multiscaling improves the model only for very short or long durations, respectively, but leads to disadvantages in modeling the other duration ranges. In contrast, allowing flattening on average leads to an improve-

ment for medium durations between 1 hour and 1 day without affecting other duration regimes. Overall, the new parametric form offers a flexible and performant model for consistently describing IDF relations over a wide range of durations, which has not been done before as most existing studies focus on durations longer than one hour or day and do not address the deviation from the power law for very long durations (2-5 days).

## 1 Introduction

The number of heavy precipitation events has increased significantly in Europe (Kundzewicz et al., 2006; Tank and Können, 2003) and worldwide (Hartmann et al., 2013). Such events are related to flooding and other hazards which can cause severe damage on agriculture and infrastructure (Brémond et al., 2013). The impact of extreme precipitation depends on the temporal scale of the event: Short intense convective precipitation exhibits different characteristic consequences than long-lasting, mostly stratiform, precipitation. Examples for events on different time scales of minutes to hours, days and weeks are pluvial or





flash floods (Braunsbach, May 2016), river floodings (Elbe, 2013) and ground water floodings (Leicestershire, March 2017),
respectively.

The definition of precipitation extremes is based on occurrence probability and is quantified using quantiles (return levels)
and associated occurrence probabilities, often expressed as return-periods in a stationary interpretation. Quantitative estima-
tions of quantiles and associated probabilities mostly follow one of two popular roads: (1) block-maxima and their description

with the generalized extreme value (GEV) distribution — a heavy-tailed and asymmetric distribution — or (2) threshold ex-
ceedances and a description with the generalized Pareto distribution (GPD) (e.g. Coles, 2001). Typically, annual precipitation
maxima of different time scales are used to describe extreme rainfall events. Both GEV and GPD can be used to model ex-
treme precipitation of a certain time scale. A common problem for short time scales, the scarce availability of data with high
measurement frequency, can be overcome by modeling several time scales at once. Different time scales can be represented as

durations over which the precipitation rate is aggregated and averaged. Most frequently, daily precipitation sums are reported,
but hourly or 5-minute aggregation are also common.

A common way to describe characteristics of extremes for various durations (timescales) are intensity-duration-frequency
(IDF) curves, describing the relationship between extreme precipitation *intensities*, their *duration* (timescale) and *frequency*
(occurrence probability or average return period). These relations are known since the mid 20th century (Chow, 1953) and

have become popular among hydrologists and meteorologists. In estimating these curves one tries to exploit the assumption of
a smooth intensity-frequency relationship across different durations. This helps for interpolating between durations and it can
also improve the estimation for short durations, as for those, shorter time series are often available.

Historically, a set of GEV distributions is sought individually for a set of durations (e.g. 5min, 1h, 12h, 24h, 48h, ...) leading
to quantile (return level) estimates for specified probabilities (return periods) for all durations considered. In a second step, for

a given probability a smooth function is estimated interpolating associated quantiles across durations (García-Bartual and
Schneider, 2001). However, by using a *duration-dependent extreme value distribution* (d-GEV) (see first examples in Nguyen
et al. (1998) and Menabde et al. (1999)), IDF estimation can be carried out in one step within a single model. To achieve this,
GEV parameters are defined as functions of duration. This approach prevents the crossing of quantiles across durations and is
thus considered consistent.

It is widely agreed, that precipitation intensities for given exceedance probabilities follow a power-law-like function (scaling)
across duration (Gupta and Waymire, 1990; Veneziano and Furcolo, 2002; Burlando and Rosso, 1996) with higher intensities
for short durations. For a range of very short durations ($d \leq 1\,\mathrm{h}$), the scaling assumption does not hold, because maxima from
different durations often originate from the same event. This leads to a curvature of IDF curves, where intensity no longer
follows a power-law with decreasing duration. Bougadis and Adamowski (2006) approached this issue by using two different

duration exponents for small and for other durations. A more smooth transition can be achieved by including the curvature as
a duration offset in the parameters of the GEV distribution without explicitly distinguishing between short and long durations.
Koutsoyiannis et al. (1998) used a model with five parameters to describe the complete IDF relation for different probabilities
(return periods) and across durations. The underlying idea is based on a reparametrization of the GEV and its three characteristic





parameters location $\mu$, scale $\sigma$ and shape $\xi$. While shape $\xi$ is held constant for all durations $d$, it is further assumed that ratio

between location and scale $\mu/\sigma$ remains constant across duration $d$ (for details, see Sect. 2.3).

Ulrich et al. (2020) built on the approach of Koutsoyiannis et al. (1998) and extended it to a spatial setting with covariates for the d-GEV parameters. Although usage of both consistent modeling and spatial pooling improves model performance, the need for more flexibility of the IDF curves in longer durations is emphasized and will be addressed in the present study. Therefore, we aim to look for new parametrizations of the IDF curve's duration dependence by combining the existing approaches of

multiscaling and duration offset and also extending it by a new parameter, the intensity offset.

The commonly used variant of the d-GEV with 5 parameters (Koutsoyiannis et al., 1998) might not be flexible enough for a wide range of durations from minutes to several days. A first approach for extending the d-GEV adresses the simple scaling relation. This model assumes a scaling being independent of exceedance probability (return period). However, relaxing this assumption leads to so-called multiscaling, which allows for different scaling-like behaviour for different exceedance

probabilities (return periods). This is achieved by introducing another parameter $\eta_2$, as in Eqs. 8 and 10 in Sect. 2.3. Then, the ratio between location and scale is not constant anymore. Multiscaling is found to be effective for durations longer than 1 h (Veneziano and Furcolo, 2002; Burlando and Rosso, 1996). Van de Vyver (2018) employs the multiscaling approach in a Bayesian setting. On a global scale different scaling parameters have been investigated by Courty et al. (2019). None of the named studies combine multiscaling with curvature for short durations but focus on only one of these aspects, while our study

is aiming for a combination and analysis of three different features.

In this study we compare different ways to parametrize IDF curves, including the features multiscaling and duration offset. In addition, we present a new d-GEV parameter, the intensity offset, which accounts for the deviation from power law and flattening of IDF curves for long durations. To our knowledge, this comprehensive analysis of different features has not been conducted before. Section 2 lists the data sources and introduces the different features and their modeling equations as well

as the verification methods to analyze modeling performance. In Sect. 3, the cross-validated verification results of all features are shown with respect to modeling performance of different return periods and durations. For verification we perform a case study using rainfall gauge data from a catchment in Germany. IDF curves that include all analyzed features are presented for selected stations.

## 2   Data and Methods

We use precipitation measurements in an area in and around the catchment of the river Wupper in western Germany. In order to compare different models for the d-GEV, we use the quantile skill index (QSI) introduced in Ulrich et al. (2020) within a cross-validation setting. For the resulting IDF curves we obtain confidence intervals using a bootstrap method and test their coverage with artificial data with known dependence levels in a simulation study. The data and all necessary methods are explained in the following section.



## 2.1 Station-based Precipitation Data

Precipitation sums for minute, hour and day are provided by *Wupperverband* and the *German Meteorological Service*. Rain gauges are located in and around the catchment of the river Wupper in North-Rhine Westphalia, Germany. 100 stations are used in total. Data from stations with a distance below 250 m are combined into a single station data set, resulting in 92 grouped stations. Moreover, when two stations with different measuring frequencies are grouped together, only the observations with the higher frequency are used for the analysis. For example, when grouping two stations with hourly and daily data, respectively, the aggregation level for two days is obtained from the hourly data for years with existing hourly observations. Years with more than 10% of missing values are disregarded. Some years contain measurement artifacts, where identical rainfall values were repeated over several time steps. After consulting the data maintainers, these years are removed before the analysis. The data exhibits heterogeneity in terms of its temporal frequency and the length of the resulting time series. Figure 1 presents the availability of data over time (left) and space (right) for 3 possible temporal resolutions. More specifically, minutely data is available since 1968, whereas daily records range back to 1893.

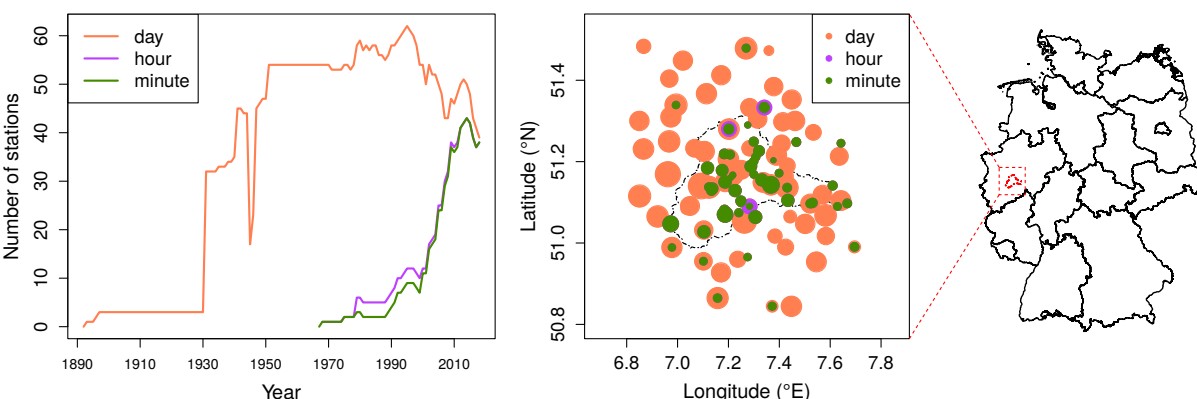

**Figure 1.** Left: Number of stations by temporal resolution. Right: Station location (circles) with data availability (circle area size) by temporal resolution (color). The red line within the map of Germany indicates the Wupper Catchment boundary.

Time series for different durations are obtained from an accumulation over a sample of durations

$$d \in \{1\,\text{min}, 4\,\text{min}, 8\,\text{min}, 16\,\text{min}, 32\,\text{min}, 1\,\text{h}, 2\,\text{h}, 4\,\text{h}, 8\,\text{h}, 16\,\text{h}, 24\,\text{h}, 48\,\text{h}, 72\,\text{h}, 96\,\text{h}, 120\,\text{h}\} \tag{1}$$

respectively, as done by Ulrich et al. (2020). In the following, numerical values are presented in hours. We use only the annual maxima of these time series to model the distribution of extreme precipitation. The original precipitation time series are accumulated with the R package `IDF` (Ulrich et al., 2021). The data set with annual maxima can be found online as supplementary material to this study (Fauer et al., 2021).





The set of durations $d$ is chosen such that small durations are presented with smaller increments than larger durations. In a simulation study we tested whether using a different set of durations with stronger focus on short durations affects the results, but no such effect could be found (see the Appendix Sect. C).

## 2.2 Generalized Extreme Value Distribution

One of the most prominent ideas of extreme value statistics is based on the Fisher-Tippet-Gnedenko theorem, which states that under suitable assumptions maxima drawn from sufficiently large blocks follow one of three distributions. These distributions differ in their tail behaviour. The GEV distribution comprises all three cases in one parametric family and is widely used in extreme precipitation analysis (Coles, 2001):

$$G(z) = \exp\left\{ - \left[1 + \xi\left(\frac{z - \mu}{\sigma}\right)\right]^{-1/\xi} \right\}. \tag{2}$$

Here, the non-exceedance probability $G$ for precipitation intensity $z$ is depending on the parameters location $\mu$, scale $\sigma > 0$ and shape $\xi \neq 0$ with $z$ restricted to $1 + \xi(z - \mu)/\sigma > 0$. The non-exceedance probability can be expressed as a return period, e.g. for an annual block size $T(z) = 1/(1 - G(z))$ years. Consequently, return values for a given non-exceedance probability $0 < p < 1$ can be calculated by solving Eq. (2) for $z$,

$$z = ([-\log\{p\}]^{-\xi} - 1)\frac{\sigma}{\xi} + \mu. \tag{3}$$

Water management authorities and other institutions rely on return values for different durations. However, the GEV distribution in the form of Eq. (2) is limited to one selected duration at a time. One way to account for that need is to model each duration separately and then, in an independent second step, interpolate the resulting quantiles (return levels) across duration $d$, as done in the KOSTRA atlas (DWD, 2017) of the German meteorological service (DWD). One huge disadvantage of this method is that quantile-crossing can occur, meaning that quantiles (intensities) associated with smaller exceedance probabilities can have higher values than quantiles from larger exceedance probabilities in some duration regimes. To solve this problem, Nguyen et al. (1998); Koutsoyiannis et al. (1998); Menabde et al. (1999) proposed a distribution with parameters depending on duration $d$; there is thus only one single model required to obtain consistent (i.e. non-crossing) duration dependent quantiles (return values). Another advantage is the involvement of data from neighbouring durations in the estimation of GEV parameters: For the modeling of short duration rainfall, often less data is available than for longer durations $d \geq 24$ (one day). Thus, in this setting, information from long durations has the potential to increase modeling performance for short durations as well.

## 2.3 Duration Dependence

There are multiple empirical formulations for the relationship between intensity $z$ and duration $d$. Koutsoyiannis et al. (1998) proposed a general form with five parameters for IDF curves. Therefore, a reparametrization and extention of the GEV is





needed with

$$\sigma(d) = \sigma_0(d+\theta)^{-\eta} \tag{4}$$

$$\mu(d) = \tilde{\mu}\sigma(d) \tag{5}$$

$$\xi = \text{const.} \tag{6}$$

$$G(z) = \exp\left\{-\left[1+\xi\left(\frac{z}{\sigma_0(d+\theta)^{-\eta}} - \tilde{\mu}\right)\right]^{-1/\xi}\right\}. \tag{7}$$

Here, $\tilde{\mu}$ is the re-scaled location parameter, $\theta$ is the duration offset and $\eta$ the duration exponent. Scale $\sigma$ follows a two-parameter power-law (scaling relation) of duration $d$, with scale offset $\sigma_0$ being constant for all durations. For $d \gg \theta$, it is justified to disable the duration offset feature by setting duration offset $\theta = 0$. The resulting IDF curves (Fig. 2a) have two main features: (1) The curves follow a power-law for a wide range of durations $d > 1$ (one hour). The power-law exponent (slope

in a double-logarithmic plot) is described by duration exponent $\eta$ and is equal for all probabilities. (2) The deviation from the power-law (or curvature) for $d < 1$ (one hour) is described by the duration offset $\theta$.

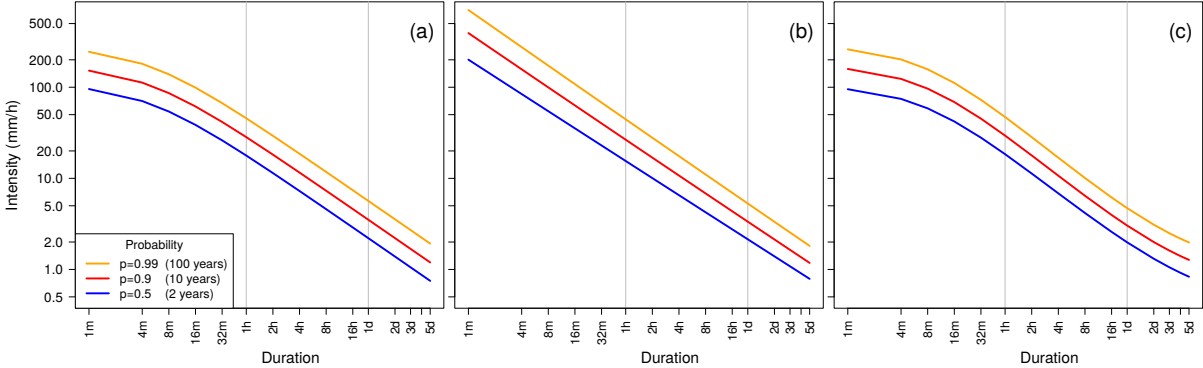

**Figure 2. IDF curve examples:** Visualization of different IDF curve features. (**a**) Curvature for short durations. (**b**) Multiscaling. (**c**) Curvature for short durations, flattening for long durations and multiscaling.

When applying the GEV separately to every duration out of a set of durations and interpolating in a second independent modeling step, the number of parameters equals three GEV parameters times the number of selected durations plus at least three parameters for interpolating every quantile. For the set of durations chosen here and for evaluating five quantiles, this

implies estimating $15 \times 3 + 5 \times 3 = 60$ parameters. For the consistent approach, estimation is reduced to only five d-GEV parameters: $\tilde{\mu}, \sigma_0, \xi, \theta$ and $\eta$. A smaller parameter set is less likely to over-fit the data and enables us to better investigate underlying physical processes.

Models can be further improved by adding the multiscaling feature (Gupta and Waymire, 1990; Burlando and Rosso, 1996; Van de Vyver, 2018), which introduces different slopes, depending on the quantile (or associated probability). Figure 2b)

presents how this feature affects the IDF curves. In order to highlight the effect of the multiscaling, we set $\theta = 0$, resulting in no deviation from a power-law (curvature) for short durations. The multiscaling feature is added at the cost of one additional





parameter $\eta_2$ (second duration exponent) in the model

$$\sigma(d) = \sigma_0 d^{-(\eta+\eta_2)} \tag{8}$$

$$\mu(d) = \tilde{\mu}\sigma_0 d^{-\eta}. \tag{9}$$

Using only the curvature feature, Ulrich et al. (2020) reported decreasing performance in consistent modeling for longer durations $d \geq 24$ (one day), compared to using separate GEV models for each duration. Our attempt aims for more flexibility of the IDF curve in the long-duration regime. Therefore, we combine both features, curvature (duration offset) and multiscaling (second duration exponent), and add a new parameter $\tau$, the intensity offset, which allows for a slower decrease of intensity for very long durations $d \gg 24$ (one day) (Fig. 2c). This effect will be called *flattening* in the following:

$$\sigma(d) = \sigma_0 (d+\theta)^{-(\eta+\eta_2)} + \tau, \tag{10}$$

$$\mu(d) = \tilde{\mu}(\sigma_0(d+\theta)^{-\eta} + \tau). \tag{11}$$

Summarizing this chapter, three different features were presented: (1) Curvature, described by the duration offset parameter $\theta$, (2) multiscaling, using a second duration exponent $\eta_2$ and (3), flattening with the intensity offset $\tau$ which is introduced in this study. In the following, we use the notation $\text{IDF}_c$ for a model including only the curvature feature, i.e. $\eta_2 = 0$ and $\tau = 0$, $\text{IDF}_m$

for a model including only the multiscaling feature, i.e. $\theta = 0$ and $\tau = 0$ and $\text{IDF}_f$ for a model including only the flattening feature, i.e. $\theta = 0$ and $\eta_2 = 0$. Feature combinations are denoted as, e.g. $\text{IDF}_{cmf}$ for all features. The plain duration dependent model without curvature, multiscaling and flattening is denoted as IDF.

### 2.4 Parameter Estimation

Parameters of the d-GEV distribution are estimated by maximising the likelihood (Maximum-Likelihood Estimation, MLE)

under the assumption of independent annual maxima. Jurado et al. (2020) showed, that independence is a reasonable assumption in many cases, especially for long durations. Their study was performed on an earlier version of the same data set that was used in this study. Moreover, Rust (2009) showed that in strongly dependent time series convergence towards the GEV distribution is slower. But, assuming an appropriate choice of model, dependence does not play a large role. We use the negative log-likelihood $L$ to avoid using products of small numbers

$$L(\tilde{\mu}, \sigma_0, \xi, \theta, \eta, \eta_2, \tau \mid Z) = \sum_{n=1}^{N} \sum_{d \in D} -\log\left(G(z_{n,d}, d \mid \tilde{\mu}, \sigma_0, \xi, \theta, \eta, \eta_2, \tau)\right), \tag{12}$$

with the number of data points $N$, the duration set $D$, data points for each duration $z_{n,d}$ and the characteristic parameters of the modified d-GEV $\tilde{\mu}, \sigma_0, \xi, \theta, \eta, \eta_2, \tau$. The sum of $L$ over all data points $Z$ is minimized with R's `optim()` function (R Core Team, 2020). The R package `IDF` (Ulrich et al., 2021) which was already used for accumulation also provides functions for fitting and plotting IDF curves. Its functionality was extended in the context of this study and now provides options for both

multiscaling and flattening in IDF curves.





Finding reasonable initial values for d-GEV parameters in the optimization process was a major challenge during parameter estimation, because optimization stability strongly depends on the choice of initial values. Details about this procedure can be found in the Appendix Sect. A.

## 2.5 Quantile Skill Index

To compare different models, we require a verification measure. In an extreme value setting, the quantile score (QS) (Koenker and Machado, 1999; Bentzien and Friederichs, 2014) describes how well a modeled quantile $q$ represents the occurrence in the data $z$ for a given probability $p$

$$QS(p) = \sum_n^N \rho_p(z_n - q),\tag{13}$$

with a small score indicating a good model. Here, $\rho_p$ is the tilted absolute value function, also known as the so-called check
function. For high non-exceedance probabilities $p$ it leads to a strong penalty for data points, that are still higher than the modeled quantile $(z_n > q)$

$$\rho_p(u) = \begin{cases} pu & ,u > 0 \\ (p-1)u & ,u \leq 0 \end{cases}\tag{14}$$

with $u = z_n - q$. Using this approach, the QS allows for detailed verification for each probability $p$ and duration $d$ separately, by predicting a quantile intensity for a given $p$ and $d$ and comparing it with data points $z_{n,d}$ of duration $d$.

The quantile skill score (QSS) compares a new model $M$ with a reference $R$

$$QSS_{M|R} = 1 - QS_M/QS_R.\tag{15}$$

The QSS takes values $-\infty < QSS \leq 1$ with $QSS = 1$ for a perfect model. Positive values $QSS_{M|R} > 0$ are associated with an improvement of $M$ over $R$. In case the model $M$ is outperformed by the reference $R$, the resulting QSS is negative $QSS_{M|R} < 0$. In this case, its value is not easily interpretable. This issue is acknowledged by the quantile skill index (QSI)
suggested by Ulrich et al. (2020): in case of $QSS_{M|R} < 0$, reference $R$ and model $M$ are exchanged and $-QSS_{R|M}$ is used for negative values of the QSI

$$QSI = \begin{cases} 1 - QS_M/QS_R & ,QS_M \leq QS_R \\ QS_R/QS_M - 1 & ,QS_M > QS_R. \end{cases}\tag{16}$$

The QSI has a symmetric range and indicates either (1) a good skill over the reference when leaning clearly towards 1, (2) little or no skill when being close to 0, or (3) worse performance than the reference when leaning clearly towards -1.

In this study, the quantile score was calculated in a cross-validation setting. For each station, the available number of annual maxima is split in blocks of three years, with one block being the testing data set and the remaining blocks being used for





training the model. All $n_{\mathrm{cv}}$ testing data sets are used once to calculate the QS. The cross-validated QS is obtained by averaging the score of all cross-validation steps

$$QS^{\mathrm{cv}} = \frac{1}{n_{\mathrm{cv}}} \sum_{i=1}^{n_{\mathrm{cv}}} QS_i. \tag{17}$$

Then, the QSI is derived from the averaged cross-validated QS of the model, $QS_M^{\mathrm{cv}}$, and the averaged cross-validated QS of the reference, $QS_R^{\mathrm{cv}}$, according to Eq. (16). If a year was assigned to the training or testing data set, then all available accumulation durations are used for training or testing, respectively to avoid dependence between test and validation set.

    In order to compare individual model features, we will use the mentioned models without this specific feature as a reference in the following.

## 2.6 Bootstrapping and Coverage

To provide an estimate of the uncertainty of the quantile estimates in IDF curves, we obtain 95% confidence intervals using a bootstrapping method. To account for dependence between annual maxima of different durations we apply the ordinary non-parametric bootstrap percentile method as follows: First, we create a sample of years of size $n$ by drawing with replacement from the $n$ years for which data are available for a certain station. That results in always using all annual maxima for a certain
year, i.e. for all durations, collectively. We expect that sampling in this way maintains the dependence structure of the data. We then estimate the parameters of the d-GEV, from which we calculate the quantiles of the distribution. We obtain a distribution of the estimated quantiles by repeating this process 500 times. We finally estimate the upper and lower bounds of the 95% confidence interval, using the empirical 0.025 and 0.975 quantiles of the distribution of estimated quantiles.

    We conduct a simulation study to examine whether the derived confidence intervals provide reasonable coverage despite
the dependence between the annual maxima of different durations. To this end, we use a sample of size $n = 50$ years and 500 samples of data with known dependence between durations. We obtain a sample with no dependence between durations, by drawing random values from a d-GEV distribution. Further details on the simulated data can be found in Appendix Sect. B. In a second case and to obtain data with dependence between durations, we use the R-Package `SpatialExtremes` to simulate values from a Brown-Resnick simple max-stable process with known dependence parameters. We transform the simulated data
from having unit-Fréchet margins to d-GEV margins with the chosen parameters, as done by Jurado et al. (2020) and adjust them to the hourly scale used in this study. We use the range and smooth parameter $(\rho, \alpha) \in \{(1, 0.2), (120, 1), (60, 1)\}$ for weak dependence, strong dependence, and a dependence found for Wupper catchment (Jurado et al., 2020), respectively. Then, for the artificial data, confidence intervals are obtained by bootstrapping with 500 repetitions. The ratio of samples, where the confidence intervals cover the true intensity $z$ and the total number of stations is called the coverage. It can be calculated for
each duration and probability separately.



## 3 Results

Results are presented in the following order: (1) Modeling performance is verified with the QSI for the three different IDF curve features: curvature, multiscaling and flattening. (2) IDF curves with all three features are shown for two rain gauges. Curves are presented with a 95% confidence interval, created by a bootstrapping method. (3) The trustworthiness of this bootstrapping
method applied to the new model with all three features is investigated with a coverage analysis, based on simulated data.

### 3.1 Model Validation

The QSI is used to compare the quantile score of a model with that of a reference. In order to specifically investigate the influence of a single model feature, we use these features in a model and compare with a reference without this specific feature, e.g. $QSI_{\mathrm{IDF}_c|\mathrm{IDF}}$ gives the performance for a model including only curvature against the plain reference without curvature or
$QSI_{\mathrm{IDF}_{cmf}|\mathrm{IDF}_{mf}}$ gives the performance for the full model including curvature, multiscaling and flattening against a reference with multiscaling and flattening and without curvature (see Sect. 2.3).

Figure 3 shows the QSI for an IDF model including each of the three features curvature, multiscaling and flattening (columns) combined with no other, one other, or both other features (rows) against a reference model which differs only in the one feature under investigation (labels on top of the columns). Models and references are listed in Table E1 in Appendix Sect. E. This way,
the potential performance of each feature, e.g. *curvature* is analyzed and is denoted as e.g. *curvature skill*. QSI values between -0.05 and 0.05 are considered as indicator of no relevant difference between model performances.

The curvature (duration offset $\theta \neq 0$) for short durations can be explained by a stronger connection between annual maxima of different durations, which tend to origin from the same event. Usually, the most intense phase of a heavy precipitation event lasts for several minutes and aggregated maxima do not differ much on this scale. Based on this idea, curvature influences the
IDF curve's shape only for very short durations below one hour (Fig. 2a). The consistently positive QSI values for $d = 1/60$ (one min) for the curvature skill support this theory. These results show that this duration regime $d = 1/60$ is much better modeled with the curvature, compared to models without this feature. However, the slope for medium durations, described by the duration-exponent (see Fig. 2b) is steeper when using curvature, compared to models that do not use curvature. So, for medium and long durations, models perform equally well or worse when curvature is used than reference models without
curvature in most cases on average (see blue regimes in Fig. 3a). In absence of multiscaling (rows 1 and 3), further performance increase could be found for durations between 8 hours and 5 days.

Multiscaling allows for different slopes of different $p$-quantiles on a double-logarithmic scale. Figure 3b shows that this feature increases modeling performance mainly for long, but also for some sub-hourly durations, when estimating quantiles for small non-exceedance probabilities. Not using curvature enables a small multiscaling skill gain for sub-hourly durations
(rows 1 and 3 in Fig. 3b). An explanation could be that multiscaling tends to let IDF curves associated with different return periods diverge for short durations and converge for long durations. This behavior might interfere with the duration offset's introduction of curvature in short durations. Furthermore, the presence of curvature leads to a slightly smaller skill increase for durations longer than 16 hours (rows 2 and 4 in Fig. 3b). This effect agrees with the results from the curvature skill verification



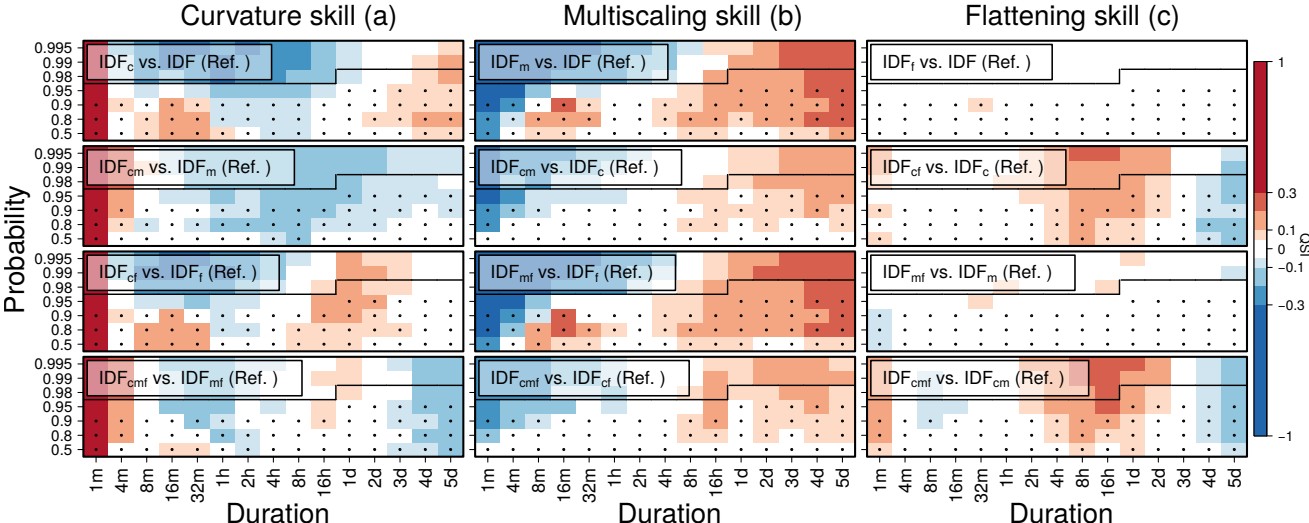

**Figure 3. Quantile skill index** (QSI) of the three features (columns) for 4 different cases (rows). Column titles indicate feature switched on in the model and switched off in the reference. Transparent labels in the panels indicate which model and reference is used, see also Table E1. Dots show whether the average length of time series over all stations is longer than the return-period $T = 1/(1-p)$ (here shown as probability $p$) and indicate verification trustworthyness. Black lines are derived from the number of years of the station with the longest time series. Verification for rare events (upper part of each panel) above the black line has to be treated carefully, because the data does not cover this time period. For this verification plot, only stations that provide data on a minutely scale were used.

(Fig. 3a), where it was shown that curvature improves modeling performance only for very short durations and has no or a
negative effect on medium and long durations.

The intensity offset $\tau$ is a new feature, first introduced in this study, which addresses the empirically observed slower decrease in intensity for very long durations, called flattening. In the case where curvature is enabled in both model and reference (rows 2 and 4 in Fig. 3c), the flattening feature improves modeling performance slightly for the shortest duration of 1 minute and strongly for medium durations between 2 hours and 1 day. Here, the flattening might compensate for the loss in skill that we
observe for medium durations for models with curvature. In these cases, there is a slight loss in skill for very long durations. In cases where curvature is not used, flattening is not needed as it provides no clear skill. An explanation for flattening of the IDF curves in long durations could be seasonal effects, with annual maxima of short or long durations occurring more often in summer or winter months, respectively. These effects are currently under further investigation.

When modeling only durations $d \geq 1$ (one hour) of all available stations, the models are rather indifferent towards parametriza-
tion (Fig. 4). Here, multiscaling and flattening show some skill improvements for long and medium durations, respectively, similar to that in Fig. 3, but to a much smaller extent, compared to a data set which uses the whole range of durations from 1 minute to 5 days for both training and testing (Fig. 3).





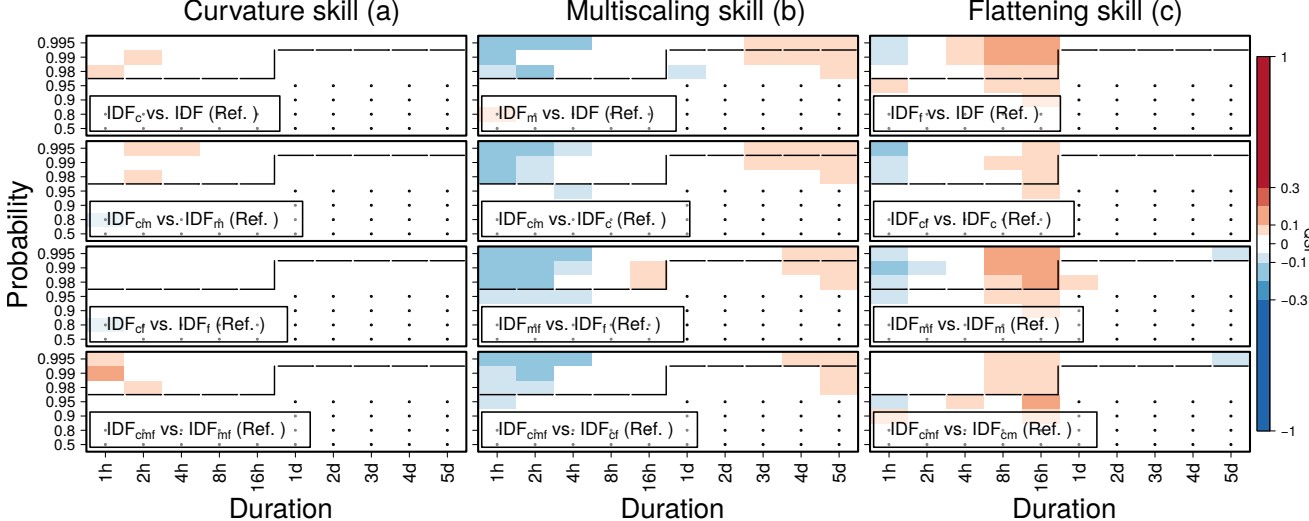

**Figure 4. Quantile skill index for data with hourly resolution:** The visualization scheme follows that of Fig. 3. Here, all models were trained and tested for durations d≥1. Here, all stations were considered, regardless of temporal resolution.

We conclude that the choice of parameters depends on the study purpose. When focusing on long ranges of durations, we recommend the usage of features like curvature, multiscaling and flattening. If focus lies on long durations or the data does not provide a sub-hourly resolution, simple scaling models might be sufficient. These recommendations are further elaborated in the discussion in Sect. 4.

### 3.2 IDF Curves

Figure 5 shows IDF curves for the stations Bever and Buchenhofen, where a long precipitation series is available (51 years with minutely resolution and 76 years with daily resolution in Bever and 19 and 77 years in Buchenhofen, respectively). The difference in available years for different durations have an impact on the width of 95%-confidence intervals, with uncertainty being larger when less data is avaiable. Noticeably, confidence intervals for for both stations for $p = 0.99$ and $d = 1/60$ have a wide range over more than 100 mm/h. Considering, that the 100-year return level was observed in neither of both stations, a wide confidence interval range was expected. For $p \leq 0.8$ in Bever, the confidence intervals remain narrow, even on a minutely scale.

### 3.3 Coverage

Confidence intervals in Fig. 5 are obtained from a bootstrapping procedure. In the formulation of the likelihood we assume the maxima of different durations to be independent. This assumption might not be justified especially for short durations (see Jurado et al., 2020) and thus this dependence must be taken into account when estimating uncertainties. Disregarding the dependence would result in an underestimation of the uncertainty. To account for this effect, all annual maxima of each



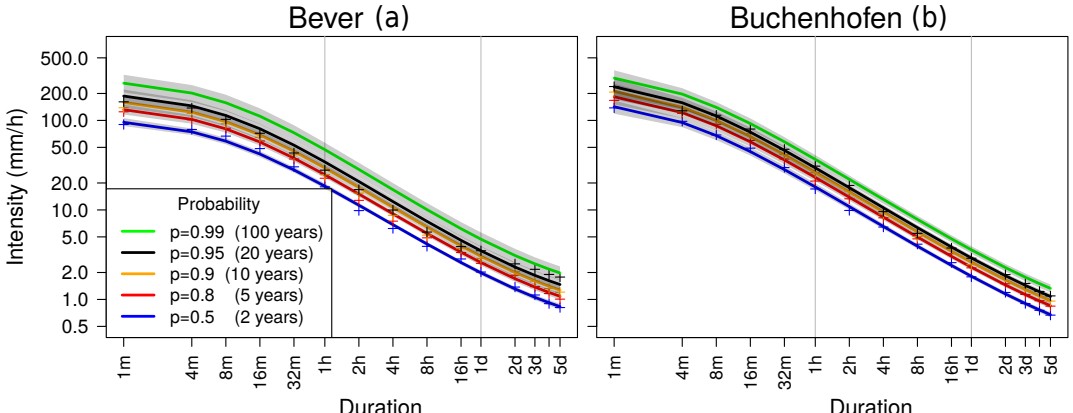

**Figure 5. IDF curves for two example stations within the Wupper catchment:** Empirical quantile estimates are denoted as "+". Confidence intervals are obtained from a bootstrapping procedure.

year — for all considered durations — are always included jointly into a bootstrapping sample. We assume that this procedure preserves the dependence structure between durations. To investigate this assumption, we calculate the coverage of simulated data (see Appendix Sect. B) from (1) a d-GEV distribution without dependence, and a Brown-Resnick max-stable process with (2) typical dependence for Wupper station (Jurado et al., 2020) and (3) rather weak and (4) strong dependence between durations (Fig. 6). In the first case without any dependence, the displayed coverage does completely agree with the 95% confidence interval, without any respect to duration or frequency (probability). When using dependence on a weak or strong level, the coverage is smaller, but still around 90%. This can be interpreted as an under-estimation of uncertainty to a small extent by the confidence intervals, in case of a high dependence. The true dependence of durations was not investigated in this study and could be lower. That said, these results suggest that bootstrapping is a suitable tool to estimate confidence intervals in the presented context.

## 4 Discussion

In this study, we show that model performance can be increased when the flattening of IDF curves in the long-duration regime is taken into account. We assume that this behavior arises from seasonal effects. That means, annual maxima of different durations may not follow the same scaling process. However, this topic is currently under further investigation.

The analyzed features — curvature, multiscaling and flattening — were seen in the results to have a different impact on modeling performance, depending on duration and return period. All features are able to improve the model for certain regimes, but depending on the problem that is approached, features should be chosen accordingly. If focus lies on small time scales of minutes, the curvature skill is important for a good modeling result. When curvature is used and medium to long time scales are also of importance the flattening feature should be used. This helps to compensate for the deterioration due to curvature



**Figure 6. Bootstrapping coverage:** Using a Brown-Resnick max-stable process, the coverage was determined in order to investigate the reliability of 95% confidence intervals from bootstrapping. Three different levels of dependence were used.





over longer durations. Multiscaling is a good choice, if a loss in skill for short durations can be accepted in exchange for
simultaneous improvement at long durations, regardless of which other features are requested.

The skills of features depend on another feature's presence. This dependence is strongest for the flattening, which can only
improve the model when curvature is used. The modeling performance of curvature depends less on the presence of other
features. The same applies to the multiscaling feature.

These suggestions hold for models that are supposed to cover a wide range of time scales from minutes to days. For data
with hourly or more coarse temporal resolution, the skill gain from using the features is much smaller. Here, flattening can
improve the model slightly on a daily time scale and multiscaling only improves modeling long durations a little bit, but leads
to a slight reduction in skill for the hourly time scale.

Additional parameters give the model more flexibility. Including $\tau$ in the model allows to reduce deviations between model
and data points particularly for long durations. This, in turn, opens the possibility to vary the remaining parameters such
that deviations between model and data points can be reduced in other (e.g. short) duration regimes. Vice cersa, this holds
for including $\theta$. This way, a parameter that changes the curve in long durations can increase modeling performance in other
durations and even slightly decrease model performance in long durations in certain cases. In Fig. 7, IDF curves for two models
are compared for a chosen station (Bever). The model that includes flattening ($\mathrm{IDF_{cmf}}$) is able to follow the empirical quantiles
in long durations as good as the model without flattening ($\mathrm{IDF_{cm}}$). However, flattening gives the model the opportunity to better
follow the empirical quantiles in medium durations between 4 h and 1 day which accords with the results in Fig. 3c.

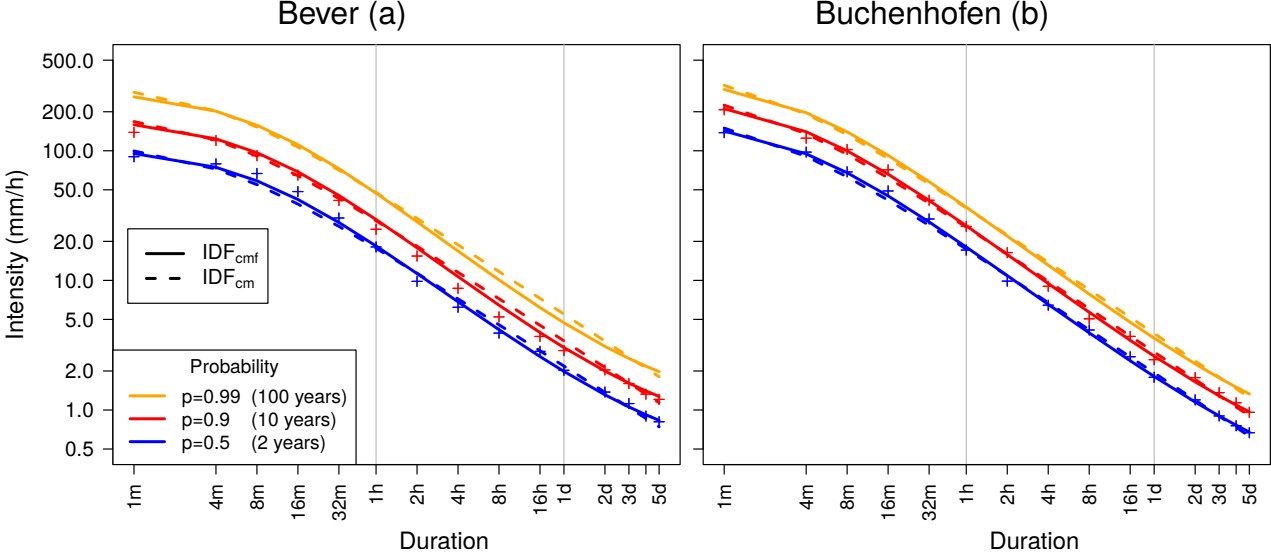

**Figure 7. IDF curve for Bever:** A comparison of a model with flattening ($\mathrm{IDF_{cmf}}$) and a model without flattening ($\mathrm{IDF_{cm}}$). Empirical quantile
estimates are denoted as "+".





The parametric form of the IDF relation is based on three modifications to a simple power-law which are motivated by our understanding of the rainfall process: *curvature* (e.g. Koutsoyiannis et al., 1998) for small durations addressing limits to rainfall intensity, *multiscaling* (e.g. Van de Vyver, 2018) taking care of a varying scaling behaviour for events of different strength, and *flattening* (suggested here) resulting from a mixing of convectively and stratiformly generated precipitation extremes from

different seasons in the climate regime under study. Our contribution is to combine these modifications into a flexible parametric form capable to describe various effects related to rainfall processes. The resulting model is based on empirical grounds and it can be shown to improve the description over simple power-law models. Other modifications are possible. To our knowledge there is no theoretical justification for these forms which can be derived from first principles for rainfall processes. Since our results might apply only to the geographical region under investigation, further studies are necessary to find out whether the

found model performances of the different features are generally applicable.

## 5 Summary and Outlook

The aim of this study is to compare and suggest new parametric forms of consistent IDF curves that are applicable to a large range of durations from minutes to several days and therefore cover events from short-lived convective storms to long-lasting synoptic events. The dependence on duration is implemented in location and scale parameter and allows for three features:

curvature, multiscaling and flattening. The analysis of these features enables us to understand more about underlying physical effects and provides more flexible IDF curves that are suitable for a wide range of durations. The results of our simulation study show that we are able to provide reasonable estimates of uncertainty using bootstrap, also with regard to dependence between durations.

Our findings agree with Veneziano and Furcolo (2002), who found simple scaling being adequate in modeling short dura-

tions and multiscaling for long durations. Moreover, our conclusion that curvature improves the modeling of short durations indirectly agrees with Bougadis and Adamowski (2006) who used different slopes for durations longer or shorter than 1 hour, respectively and concluded that linear scaling does not hold for small durations.

In future studies we plan to include spatial covariates into the estimation of the newly proposed d-GEV parameters, including intensity offset, in order to use data from different locations more efficiently. Also, atmospheric large-scale covariates shall be

investigated to study the change in characteristics of extreme precipitation due to climate change. Moreover, the origin of flattening in annual maxima for long durations is currently investigated in more detail.

The analysis of performance shows that the new parametric form of the duration-dependent GEV suggested here together with the bootstrap based confidence intervals offer a consistent, flexible and powerful approach to describe intensity duration frequency (IDF) relationships for various applications in hydrology, meteorology and other fields.

*Code and data availability.* Parts of the rainfall data are freely available from the German Weather Service (DWD) (DWD, n.d.). Annual precipitation maxima of stations from Wupperverband and DWD for 15 different durations are available as supplementary material to this





study (Fauer et al., 2021). The d-GEV analysis is based on the R-package *IDF*, available on CRAN (https://CRAN.R-project.org/package=IDF) (Ulrich et al., 2020, 2021; R Core Team, 2020).

### Appendix A: Initial Values

The estimation of d-GEV parameters is conducted with the R base `optim()` function (R Core Team, 2020) and the `Nelder-Mead` method. Quality of the fitted model depends on the initial values, passed to the function. Each optimization $i$ was repeated $m$ times with different initial values that are derived through different techniques. The sets of initial functions $s_i \in \{\tilde{\mu}', \sigma_0', \xi', \theta', \eta', \eta_2', \tau'\}$ are collected as suggestions and the individual parameters are named with version indices (`v1, v2, ...`). All suggestions were used as initial values in the model subsequently and the suggestion which lead to the smallest negative log likelihood was selected. Table A1 gives an overview of the combinations of initial values.

All initial-value techniques were based on the same first step. An individual GEV distribution was fitted to each duration $d$ separately, with moment estimators as initial values (Coles, 2001), and the three GEV parameters location $\mu$, scale $\sigma$ and shape $\xi$ were stored for each duration. In the next step, a function was fitted to each of the parameters with respect to the duration. Since we assumed no dependence of the shape parameter on the duration, we chose $\xi_{v1}' = \text{median}(\xi)$ for all suggestions. According to Eqs. (4) and (5), we fit $\log(\sigma(d))$ and $\log(\mu(d))$ as a function of $\log(d)$ in a linear regression with a simple slope and y-intercept:

$$\log(\sigma(d)) \sim -(\eta + \eta_2)\log(d) + \log(\sigma_0) \tag{A1}$$

$$\log(\mu(d)) \sim -\eta\log(d) + (\log(\sigma_0) + \log(\tilde{\mu})) \tag{A2}$$

with given $\sigma(d)$, $\mu(d)$, $d$. From this fit, we extract $\sigma_{0,v1}' = \exp(\log(\sigma_0))$ (Eq. (A1)), $\tilde{\mu}_{v1}' = \tilde{\mu}$ (combine Eq. (A1) and A2), $\eta_{v1}' = \eta$ and $\eta_{2,v1}' = \eta_2$. For the most simple suggestion of initial parameters, we chose $\theta_{v1}' = 0$ and $\tau_{v1}' = 0$. In the next steps, we further elaborate the ways of finding good initial values. Version 2 of duration exponents $\eta_{v2}'$ and $\eta_{2,v2}'$ are found, using only d≥1 h, because the slope is mainly characterized by this duration regime. This is the second set of suggestions for initial values, together with version 1 of the other parameters.

Another initial duration offset $\theta_{v2}'$ can be estimated by fitting a non-linear-squares regression (R function `nls`):

$$\log(\sigma(d)) \sim -(\eta + \eta_2)\log(d + \theta) + \log(\sigma_0) \tag{A3}$$

$$\log(\mu(d)) \sim -\eta\log(d + \theta) + \log(\sigma_0\tilde{\mu}) \tag{A4}$$

with given $\sigma(d)$, $\mu(d)$, $d$. The mean of both estimates for $\theta$ was used for $\theta_{v2}'$. These functions are less stable and provide worse initial values for $\sigma_0'$ and $\tilde{\mu}'$ than Eq. (A1) and A2. That is why we use them only for estimating initial $\theta_{v2}'$ in the `nls` function and not for estimating initial $\tilde{\mu}$, $\sigma_0$, $\eta_1$ or $\eta_2$. The new initial estimate $\theta_{v2}'$ was combined with $\eta_{v1}'$ and $\eta_{2,v1}'$ in one set (suggestion 3) and with $\eta_{v2}'$ and $\eta_{2,v2}'$ in another set (suggestion 4).





**Table A1.** Overview of initial value combinations (suggestions). The initial values for the parameters $\tilde{\mu}'_{\text{v1}}$, $\sigma'_{0,\text{v1}}$ and $\xi'_{\text{v1}}$ are the same in all combinations.

| Suggestion number | Version | | | |
|---|---|---|---|---|
| | $\theta'$ | $\eta'_1$ | $\eta'_2$ | $\tau'$ |
| 1: | v1 | v1 | v1 | v1 |
| 2: | v1 | v2 | v2 | v1 |
| 3: | v2 | v1 | v1 | v1 |
| 4: | v2 | v2 | v2 | v1 |
| 5: | v1 | v1 | v1 | v2 |
| 6: | v1 | v2 | v2 | v2 |
| 7: | v2 | v1 | v1 | v2 |
| 8: | v2 | v2 | v2 | v2 |

The same `nls` function was used for an estimation of initial $\tau'_{\text{v2}}$, taking only d≥1 h and no duration offset (curvature):

$$\log(\sigma(d)) \sim \log(\sigma_0 d^{-(\eta+\eta_2)} + \tau) \tag{A5}$$

$$\log(\mu(d)) \sim \log(\sigma_0 \tilde{\mu} d^{-\eta} + \tau) \tag{A6}$$

with given $\sigma(d)$, $\mu(d)$, $d$. Again, the mean of both estimates of $\tau$ is used as $\tau'_{\text{v2}}$. To define suggestions number 5-8, this second version of $\tau'$ was combined with $\theta'_{\text{v1}}$, $\eta'_{\text{v1}}$, $\eta'_{2,\text{v1}}$ or $\theta'_{\text{v1}}$, $\eta'_{\text{v2}}$, $\eta'_{2,\text{v2}}$ or $\theta'_{\text{v2}}$, $\eta'_{\text{v1}}$, $\eta'_{2,\text{v1}}$ or $\theta'_{\text{v2}}$, $\eta'_{\text{v2}}$, $\eta'_{2,\text{v2}}$. The different combinations of initial value versions are listed again in Table A1.

## Appendix B: Simulated Data

For the coverage analysis in Sect. 3.3 and Appendix Sect. C about duration sample choice, we do not use the original data set, but simulated data from a d-GEV distribution. The simulated data is drawn, according to Eqs. (3), (10) and (11) with a random number $0 < p < 1$ and the parameters $\tilde{\mu} = 3.2$, $\sigma_0 = 5.8$, $\xi = 0.21$, $\theta = 0.089$, $\eta = 0.78$, $\eta_2 = 0.09$ and $\tau = 0.10$. These values are based on realistic parameter values from station data, fitted for the features curvature, multiscaling and flattening. When disabling one or more of the features, values all parameters will change. For the coverage analysis, the true quantile intensity $z$ can be calculated directly, using these parameters and Eqs. (3), (10) and (11).

## Appendix C: Influence of Duration Sample Choice

We investigate how the choice of durations that are used to train the model influences the model performance. Since the number of training data points is much higher for long durations $d \geq 24$ (one day), there is a possibility that these duration regimes





are over-represented in the training phase and thus, model performance is worse for short durations. To account for this effect, the model was trained twice, (1) with simulated maxima (Appendix Sect. B) and the set of aggregated durations that was used for analysis in this study (Eqs. (1) and (2)) with simulated annual maxima of a different set of durations, that focuses more on short durations (numerically in hours):


$$d_2 \in \{1, 2, 3, ..., 15, 16, 18, 20, ..., 30, 32, 33, 36, 39, ..., 57 \, \text{min}, \qquad 1, 2, 3, ..., 6, 8, 10, 12, 15, 16, 18, 21 \, \text{h}, \qquad 1, 3, 5 \, \text{d}\}.$$

This way, there is more training data for short durations available, which might shift the model's performance focus to other duration regimes. However, it is important to note, that this is only an artificial increase of available data, since the additional data points do not contain substantial new information.

The model with the new artificial training data set (Eq. (**??**)) is now verified against the same model with the previously used artificial data set (Eq. (1)) with the following results: All QSI are below 0.05 for all durations $d$ and all quantiles $q \in \{0.5, 0.8, 0.9, 0.95, 0.98\}$ (not shown). Thus, the results do not indicate that the choice of accumulation durations significantly influences how well the model performs for certain duration regimes.

### Appendix D: Model Diagnosis

In order to evaluate whether the GEV distribution is an appropriate choice for this analysis we prepared quantile-quantile-Plots (QQ-Plots) (Fig. D1) for the stations Bever and Buchenhofen, as chosen in Sect. 3.2. While for Buchenhofen all values follow the angle bisecting line, for Bever some high empirical quantiles are outside of the confidence interval. However, compared to the number of shown data points, only very few leave the confidence intervals and we conclude that the GEV distribution is a suitable assumption in our case.

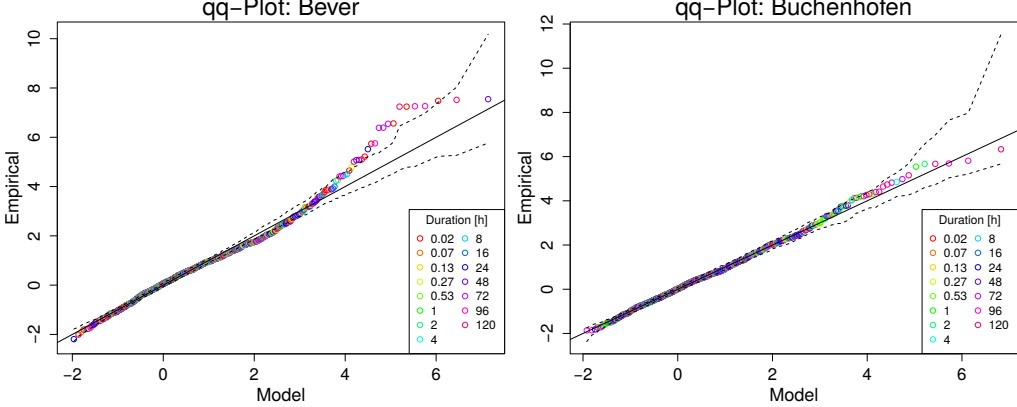

**Figure D1. QQ plots for selected stations.** Confidence intervals were obtained by simulating transformed unit-Fréchet distributed values from the model distribution and extracting a 95% interval.





**Appendix E: Overview of Reference Models for Verification**

In Sect. 3.1, feature skill is evaluated by comparing models with a reference, where the considered feature is disabled. For clarification, Table E1 lists the models and reference models that are used in Fig. 3 and 4 together with the parameter restriction to zero, if applied. These specifications refer to Eqs. (10) and (11).

**Table E1.** Overview of models and references for verification.

| Column (title) Row (number) | Model features and parameters | Reference features and parameters |
|---|---|---|
| Curvature | | |
| 1 | $IDF_c$ $\theta \neq 0, \eta_2 = 0, \tau = 0$ | IDF $\theta = 0, \eta_2 = 0, \tau = 0$ |
| 2 | $IDF_{cm}$ $\theta \neq 0, \eta_2 \neq 0, \tau = 0$ | $IDF_m$ $\theta = 0, \eta_2 \neq 0, \tau = 0$ |
| 3 | $IDF_{cf}$ $\theta \neq 0, \eta_2 = 0, \tau \neq 0$ | $IDF_f$ $\theta = 0, \eta_2 = 0, \tau \neq 0$ |
| 4 | $IDF_{cmf}$ $\theta \neq 0, \eta_2 \neq 0, \tau \neq 0$ | $IDF_{mf}$ $\theta = 0, \eta_2 \neq 0, \tau \neq 0$ |
| Multiscaling | | |
| 1 | $IDF_m$ $\theta = 0, \eta_2 \neq 0, \tau = 0$ | IDF $\theta = 0, \eta_2 = 0, \tau = 0$ |
| 2 | $IDF_{cm}$ $\theta \neq 0, \eta_2 \neq 0, \tau = 0$ | $IDF_c$ $\theta \neq 0, \eta_2 = 0, \tau = 0$ |
| 3 | $IDF_{mf}$ $\theta = 0, \eta_2 \neq 0, \tau \neq 0$ | $IDF_f$ $\theta = 0, \eta_2 = 0, \tau \neq 0$ |
| 4 | $IDF_{cmf}$ $\theta \neq 0, \eta_2 \neq 0, \tau \neq 0$ | $IDF_{cf}$ $\theta \neq 0, \eta_2 = 0, \tau \neq 0$ |
| Flattening | | |
| 1 | $IDF_f$ $\theta = 0, \eta_2 = 0, \tau \neq 0$ | IDF $\theta = 0, \eta_2 = 0, \tau = 0$ |
| 2 | $IDF_{cf}$ $\theta \neq 0, \eta_2 = 0, \tau \neq 0$ | $IDF_c$ $\theta \neq 0, \eta_2 = 0, \tau = 0$ |
| 3 | $IDF_{mf}$ $\theta = 0, \eta_2 \neq 0, \tau \neq 0$ | $IDF_m$ $\theta = 0, \eta_2 \neq 0, \tau = 0$ |
| 4 | $IDF_{cmf}$ $\theta \neq 0, \eta_2 \neq 0, \tau \neq 0$ | $IDF_{cm}$ $\theta \neq 0, \eta_2 \neq 0, \tau = 0$ |



*Author contributions.* Conceptualization: H.W.R., F.S.F. and J.U.; Data curation: F.S.F. and J.U.; Formal analysis: F.S.F.; Funding acquisi-
tion: H.W.R.; Methodology: F.S.F. and J.U.; Software: F.S.F., J.U. and O.E.J.; Supervision: H.W.R.; Validation: F.S.F. and J.U.; Visualization:
F.S.F.; Writing – original draft preparation: F.S.F.; Writing – review and editing: J.U., O.E.J., H.W.R. All authors have read and agreed to the
published version of the manuscript.

*Competing interests.* The authors declare that they have no conflict of interest.

*Acknowledgements.* We would like to thank the German Weather Service (DWD), and the Wupperverband, especially Marc Scheibel, for
maintaining the station-based rainfall gauge and providing us with data.



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
