# Peer review of "Flexible and Consistent Quantile Estimation for Intensity-Duration-Frequency Curves"

_Hydrology and Earth System Sciences, 2021_

## Author Comment (AC1)

**Reply on RC1**

Felix S. Fauer, Jana Ulrich, Oscar E. Jurado, Henning W. Rust

August 26, 2021

Dear Rasmus Benestad,

We would like to thank you for your constructive comments. In the following we will address them point by point and support them with figures and a table.

**1 Answers**

**1.** Due to the small samples of the rainfall from clouds which rain gauges with a diameter of centimetres represent, I wonder if it's justifiable to combine data from stations with a distance below 250 m into a single station data set. It's probably OK for the statistical properties, but maybe not so for the raw data - especially on short time intervals (too much random sampling noise?). At least, there ought to be a test of the robustness of the results to this assumption.

**Answer:** Thank you for this remark. We value transparency in data processing which is why we described carefully how station data was obtained. We agree that the grouping of stations has to be treated carefully. In most cases, data with different temporal resolutions are obtained from different devices. For more transparency, we present the distance between devices of the same station in Tab. 1 (in this document). It shows that the average distance between stations is much lower than 250m. We argue that the grouping should only be relevant for maxima of $d \geq 24$h (which can be derived from minutely, hourly or daily data) and on this time scale, the spatial distance should have less impact on the results for most stations.

To verify the robustness of this approach, we investigate the influence of the grouping on the resulting estimated IDF curves. For this purpose, we use for the modeling of maxima with $d \geq 24$h (1) the maxima originating from the time series with daily resolution and (2), if available, the maxima originating from the higher resolution time series to estimate the IDF curves. In both cases the time range covers all years in which minutely data is available, so that the number of used data points is the same. Exemplarily, the two IDF curves for Bevertalsperre and Buchenhofen are shown in Fig. 1. For Buchenhofen, the differences are sufficiently small. For Bever the intensities derived from daily data are smaller, which was expected, since daily sums from daily data come from a fixed time frame (e.g. 00:00 to 23:59) whereas daily sums from minutely data come from a 24-hour window that is shifted minuteby-minute and so, larger sums are captured. When measurements for daily precipitation sums exist from different devices with different temporal resolutions, we decided to keep minutely data in our study because of the flexible time window. For most other stations, IDF curves for both cases do not differ significantly. For few stations, there are differences, especially when only short records are available. We will add these remarks to the discussion.

A major advantage of this procedure is that the full potential of the duration-dependent GEV model can be used when grouping daily and minutely data. This way, the model can profit from both long daily records and short records with high temporal resolution at the same time and information can be used and transferred efficiently.

Additionally, we will correct a wrong number. The total number of used stations is 115. After the grouping process, 92 stations remain for our analysis.

**2.** Perhaps it would make the flow of the text better if there was some connection between the return value (probability) and the quantile? I thought the part on Quantile Skill Index (e.q. 13) wasn't as easy to follow as the preceding sections (how does it link to the preceding discussion on the GEV and the estimation of the parameters?).

**Answer:** Thank you for pointing out that the part about the Quantile Skill Index was not clear. Sect. 2.4 explains how the model is trained to get the best parameter estimate. Sect. 2.5 describes the verification to evaluate the model's performance and to compare it to other models. The words "quantile" and "return value" are used synonymous. In the manuscript we refer (non-exceedance) "probabilities" $p$ to the corresponding "return periods" $T$ since they can be easily converted with $T = 1/(1 - p)$. We will rephrase the introduction in this subsection and link it to the estimation of GEV parameters.

**3.** The description of the bootstrapping was a bit difficult to follow - perhaps explain it more carefully or add an illustration?

**Answer:** Thank you for this remark. We will improve the description and add a reference to Davison & Hinkley (1997) where this procedure is described in detail.

**4.** Very brief catches (e.g. minutes) of rainfall with rain gauges are expected to be subject to a large degree of sampling uncertainty, aren't they? (depending on the number of rain drops falling onto the cross section representing the measurement). Maybe this also can explain some discrepancies at the extreme short end of the scale?

**Answer:** Thank you for this remark. Minutely measurements might indeed be less accurate when only a small number of rain drops is recorded. However, for events that are identified as annual maxima we expect the rain amount to be large enough that a higher sampling uncertainty compared to larger measurement accumulation sums can be neglected. We will add this note to the discussion.

**5.** One reason why annual maxima of different durations do not follow the same scaling process could be that different rain-producing meteorological phenomena have different temporal and spatial scales. If the rainfall can be considered as a 'by-product' of different processes and conditions (e.g. convection, weather fronts, cyclones, and derechoes), then different statistics may perhaps show the true situation? But I'm still struggling to understand what the skill estimates really say.

**Answer:** Thank you for this comment. We agree that maxima for different durations might origin from processes on different spatio-temporal scales. The aim of this new model is to be more flexible and cover extremes from different processes. Especially, it is able to combine the estimation of extremes from both short, mainly convective, processes and processes on longer time scales like derechos. It would be interesting to investigate different statistics for different durations, but then we would need to have a reliable theory that justifies which statistic should be used for which duration. The quantile score (QS) is a proper scoring rule (Gneiting, 2011) and is obtained for each duration individually. Therefore, it can evaluate the model performance, independent of underlying physical processes. Its purpose is to asses how well the modeled quantiles represent the observed maxima. Skill estimates describe how well the model performs in terms of QS, compared to a reference, especially to the commonly used models IDF or $IDF_c$ or $IDF_m$ (for model description, see Tab. E1 in the manuscript).
We will improve Sect. 2.5 about the verification accordingly to describe the purpose of the skill estimates.

**6.** A new and relevant paper DOI: 10.1088/1748-9326/abd4ab suggests a simple formula for expressing IDF curves even for sites with limited data. This formula is based on more 'physical' parameters (wet-day mean precipitation and wet-day frequency), rather than the stronger reliance on the statistical/mathematical theory behind GEV. It would be interesting to compare the results presented here with this formula. It also fits in the comparison of different ways to parametrize IDF curves. At least, it could be included in the introduction and the discussion of different ways of calculating IDFs.

**Answer:** Thank you for suggesting this relevant paper. We will include it in the introduction and/or in the discussion where we are going to add a paragraph about non-stationarity, as suggested by another reviewer. We think that a reference to this paper would fit there as well.

**7.** Appendices: when describing what calculations and processing was done in this analysis, it's more elegant to use past tense rather than present tense (my subjective opinion). But mixing past and present tense makes the text inconsistent and a 'clumsy' read. Also check the references therein ('??').

**Answer:** Thank you for these two suggested improvements. We will improve both the tenses and the references in this Section.

On behalf of all authors

Felix Fauer

**2 References**

Davison, A., & Hinkley, D. (1997): Bootstrap Methods and their Application (Cambridge Series in Statistical and Probabilistic Mathematics), Cambridge: Cambridge University Press. `https://doi.org/10.1017/CBO9780511802843`

Tilmann Gneiting (2011): Quantiles as optimal point forecasts, International Journal of Forecasting, 27, 197-207. `https://doi.org/10.1016/j.ijforecast.2009.12.015.`

**3 Figures and Tables**

[Figure]

Figure 1: Comparison of IDF curves for two different ways of obtaining maxima for $d \geq 24$h. keep min: all maxima are derived from minutely data. keep day: maxima $d \geq 24$h are derived from daily data.

Table 1: Distance between devices of grouped stations. Two bold stations are chosen in the manuscript for detailed IDF curves.

| Index | Number | Station name | Distance (m) |
|---|---|---|---|
| 1 | 3 | Breckerfeld-Wengeberg | 90 |
| 2 | 16 | **Hückeswagen (Bevertalsperre)** | 101 |
| 3 | 18 | Köln-Stammheim | 49 |
| 4 | 30 | Remscheid-Lennep | 34 |
| 5 | 32 | Solingen-Hohenscheid | 22 |
| 6 | 35 | Wipperfürth-Gardeweg | 52 |
| 7 | 37 | **Wuppertal-Buchenhofen** | 217 |
| 8 | 50 | Bochum | 0 |
| 9 | 51 | Dormagen-Zons | 0 |
| 10 | 53 | Köln-Bonn | 0 |
| 11 | 54 | Essen-Bredeney | 0 |
| 12 | 64 | Reichshof-Eckenhagen | 0 |
| 13 | 65 | Neunkirchen-Seelscheid-Krawinkel | 0 |
| 14 | 66 | Lüdenscheid | 0 |
| 15 | 67 | Meinerzhagen-Redlendorf | 0 |
| 16 | 68 | Overath-Böke | 0 |
| 17 | 69 | Gevelsberg-Oberbröking | 0 |
| 18 | 72 | Leverkusen | 4 |
| 19 | 74 | Neumühle | 37 |
| 20 | 75 | Schwelm | 24 |

---

## Author Comment (AC2)

**Reply on RC2**

Felix S. Fauer, Jana Ulrich, Oscar E. Jurado, Henning W. Rust

August 27, 2021

Dear Reviewer,

We would like to thank you for your constructive comments. In the following, we will address them point by point and support them with figures.

**1    Answers**

**1.** Line # 29: could be written as Methods

**Answer:** Thank you for this suggestion. We will change this sentence.

**2.** In lines # 65-70: A few aspects of nonstationarity could be discussed here, discuss briefly the added value of this study to address nonstationarity as compared to the methods discussed in the literature (Cheng and AghaKouchak 2014; Ganguli and Coulibaly 2017).

**Answer:** Thank you for suggesting these relevant papers. We will include non-stationarity in the discussion because we agree that this is an important concept that should be considered in IDF curves. However, this study is not supposed to add value in non-stationarity, since stationarity is an assumption of our model. We plan to include non-stationarity in future studies.

**3.** Section 2.1: Lines 90 – 95: This is not very clear - why 8 stations were merged into a single station leaving 92 overall stations out of 100 stations? Which physiographic or hydrologic similarity measures were adopted for regionalization?

**Answer:** Thank you for pointing out that this is not clear. We group the stations because the model can use its full potential, when long daily records are combined with high resolution (minute) records which are often much shorter. Typically, two different measuring frequencies are obtained from different devices and in some cases, those devices are not positioned at the same site. However, in most cases those stations are very near to each other or even at the same position. Therefore, whenever two stations have a distance of less than 250m, those respective stations are merged into one station. The only measure for this procedure is the distance. For more details about the distance of the stations and a test of robustness,

see also the answer to comment 1 in RC1 (`https://hess.copernicus.org/preprints/hess-2021-334#AC1`).

We will describe that more clearly in the manuscript. Also, we will correct a mistake in the manuscript. The total number of stations is 115. The number of grouped stations is 92.

**4.** It has been shown in the literature that the Generalized Maximum likelihood method, in general, does not provide a credible estimate of the shape parameter, yielding an abrupt estimate of shape estimate (Martins and Stedinger 2000). Have your values lie within the credible limits of shape parameter range as shown in the literature, boxplots showing the range of shape parameters for different duration could help to identify this issue

**Answer:** Thank you for this remark. Since in our model we assume the shape parameter to be constant over duration, we cannot present the range of the shape parameter depending on duration. However, we present the range of the shape parameter at all used stations in Fig. 1 to investigate whether it lies within credible intervals. Almost all parameters lie within $-0.3 < \xi < 0.6$. Only for one station (Dabringhausen), an unrealistically large shape parameter was estimated which could be explained by the scarce data availability at this station (15 years).

Martins and Stedinger (2000) report that maximum likelihood estimation (MLE) in small samples can lead to unrealistic shape parameters and that for large sample sizes the RMSE of both methods become similar (Figs. 4 and 5 in their study). We argue that due to the duration-dependent GEV in our study the number of data points available for estimation is multiplied by the number of duration steps and so, sample size should be sufficiently large for using MLE.

**5.** In skill score index in lines 190-200: what M and R represent, If R denotes empirical distribution, which empirical plotting position formula was used to estimate it? Typically Gringorten's plotting position formula is in use to characterize extremes.

**Answer:** R does not denote the empirical distribution but another IDF model. The quantile skill score (QSS) compares the quantile score ($QS_M$) of a new model M with the quantile score ($QS_R$) of a reference model R. The difference between models and references are only characterized by the features (curvature, multiscaling, flattening) that are used/not used in this model or reference. So, different combinations will be compared, e.g. a model using curvature and multiscaling (IDF$_{cm}$ as model M) vs. a model using only curvature (IDF$_c$ as reference model R). Table E1 in the manuscript lists which feature combinations are used as model M and reference R. The quantile score (QS) is a propper scoring function (Gneiting, 2011) for comparing modeled quantiles to all observations. We do not calculate a difference between model quantiles and empirical quantiles. Empirical distributions are only used for visualization ("+" in Figures 5 and 7 in the manuscript).

Thank you for pointing out that this part was not clearly described. We will improve Section 2.5 accordingly.

**6.** Fig. 7: Could you please show the difference in return levels in an inset diagram with vs without flattening? How much is the percentage difference between the two statistics in order to qualify as significant?

**Answer:** We created additional panel plots that show the difference in intensity and also the ratio of both estimates between the two models (Fig. 2 in this document). In comparison with Fig. 5 from the manuscript the differences are expected to overlap with the confidence intervals, suggesting non-significance. However, these confidence intervals are mainly depending on data size and would become smaller with more data available. Nevertheless, the purpose of this visualization was to demonstrate that allowing for flattening of IDF curves in long durations has an impact on the shape of IDF curves in short durations. These figures only show two selected stations, exemplarily.

In our study, the verification of model performance for all stations was done with the Quantile Skill Index (QSI) where we could show that certain models improve IDF estimation in certain duration-regimes, depending on the use-case. Here, a QSI<0.05 is considered as irrelevant (see white regions in Figs. 3 and 4 in the manuscript).

**7.** Between lines 360-363: Any discussion on copula-based IDF estimation that claims to preserve the inherent non-linearity between intensity vs duration?

**Answer:** Thank you for this remark. Bezak et al. (2016) used copula-based IDF curves and reported that IDF curves might be sensitive to the choice of method. This is important to consider when deciding on the appropriate way to create IDF curves. We will include this reference in the manuscript.

**7.** Line # 433: few outlying events correspond to higher quantile (or at the tail of the distribution) leave the confidence intervals.

**Answer:** Thank you for this suggestion. We will change this sentence.

On behalf of all authors
Felix Fauer

**2   References**

Martins ES, Stedinger JR (2000): Generalized maximum-likelihood generalized extreme-value quantile estimators for hydrologic data. Water Resources Research, 36,737–744, `https://www.doi.org/10.1029/1999WR900330`.

Nejc Bezak, Mojca Šraj, Matjaž Mikoš (2016): Copula-based IDF curves and empirical rainfall thresholds for flash floods and rainfall-induced landslides. Journal of Hydrology, 541,272-284, `https://doi.org/10.1016/j.jhydrol.2016.02.058`

Tilmann Gneiting (2011): Quantiles as optimal point forecasts, International Journal of Forecasting, 27, 197-207. `https://doi.org/10.1016/j.ijforecast.2009.12.015`

**3  Figures and Tables**

[Figure]

Figure 1: A histogram, showing the values of shape parameters from all stations. The shape parameter is constant across durations. The median is shown with a red line.

[Figure]

Figure 2: **Upper panels:** Fig. 7 from the manuscript. **Lower panels:** difference and ratio between the two models shown above.

---

## Author Comment (AC3)

**Reply on RC3**

Felix S. Fauer, Jana Ulrich, Oscar E. Jurado, Henning W. Rust

September 21, 2021

Dear Reviewer,

We would like to thank you for your constructive comments. In the following, we will address them point by point.

**1 Answers**

**1.** How was the value of zn estimated (refer equation 13) for computing the Quantile Skill Index (QSI)? Did the authors perform any sensitivity analysis to evaluate the effect of plotting position used on QSI?

**Answer:** The value of $z_n$ is not estimated, but is directly taken from the data z. A modeled quantile is evaluated using all annual maxima $z_n$ by the quantile score. Since we do not use empirical quantiles for verification, we did not perform a sensitivity analysis for the plotting position.

**2.** The cross validation setting used needs a more clear explanation. How was the block for cross validation choosen?

**Answer:** Thank you for pointing out, that this part is unclear. For each station, the available years of data are divided into non-overlapping blocks of three consecutive years. Then, for each cross-validation step, one block of years is chosen as testing set and all the other blocks are used as training data set. For the remaining cross-validation steps this procedure is repeated with another block chosen as testing set in each step until all blocks have been used as testing sets exactly once.
We will add this explanation in chapter 2.5.

**3.** Section 2.6 on boot strapping and coverage is confusing and needs a more clear description.

**Answer:** Thank you for this remark. We will improve that chapter.

---

## Author Comment (AC4)

**Reply on RC4**

Felix S. Fauer, Jana Ulrich, Oscar E. Jurado, Henning W. Rust

September 21, 2021

Dear Reviewer,

We would like to thank you for your constructive comments. In the following, we will address them point by point.

**1 Answers**

**1.** In this study, the authors investigated the features of curvature, multi-scaling and flattening on the deviation of IDF curves under the stationary assumption. Particularly, the parameters of GEV are modelled as functions of duration. Do you consider the impacts of climate change on the IDF under a changing environment? There are studies who try to update the IDF curves considering the nonstationarity, such as Agilan and Umamahesh (2017) and Ganguli Coulibaly (2017), and Yan et al. (2021) provided a review about this topic. In a nonstationary model, the parameters are modelled as function of covariates, please make a discussion or outlook about this topic, which should be highlighted under the changing environment.

**References**

[1] Agilan, V., Umamahesh, N. V. (2017). What are the best covariates for developing non-stationary rainfall intensity-duration-frequency relationship? Advances in Water Resources, 101, 11–22.

[2] Ganguli, P., Coulibaly, P. (2017). Does nonstationarity in rainfall require nonstationary intensity-duration-frequency curves? Hydrology and Earth System Sciences, 21(12), 6461–6483.

[3] Yan L, Xiong L, Jiang C, Zhang M, Wang D, Xu C-Y. (2021) Updating intensity–duration–frequency curves for urban infrastructure design under a changing environment. WIREs Water. 2021; e1519.

**Answer:** Thank you for this comment and for providing usefull references. We will include non-stationarity in the discussion because we agree that this is an important concept that should be considered in IDF curves.

**2.** In this study, the authors just consider the GEV distribution for the deviation of IDF, I think for the study area, lognormal or Gamma distribution may also exhibit comparative or better performance. It makes sense for engineering design to try other probability distributions and compare the results.

**Answer:** In an engineering context for small finite samples, other distributions than the three distributions resulting from the Fisher-Tippett Theorem might approximate the measured values indeed better for the observed time period. However, the three types of the GEV (Gumbel, Fréchet, Weibull) are the only limiting distributins for block-maxima and thus this theorem gives a strong motivation to use these distributions for modelling block-maxima (e.g. Coles, 2001). Due to this solid theoretical justification, many studies use the GEV distribution to develop IDF curves (Papalexiou and Koutsoyiannis, 2013, Van de Vyver, 2018, Shrestha et al., 2017). Including other distributions is beyond the scope of this study.

**3.** Make a discussion about the deviation of the copula-based IDF curves.

**Answer:** Thank you for this remark. We will include a short discussion about copula-based IDF curves in the manuscript.

**4.** For Figure 3, the Quantile skill index is difficult to understand, please make a clearer legend for potential readers.

**Answer:** Thank you for pointing out that this figure is difficult to understand. We agree that it is a complex figure but we were not able to find an easier visualization so far. The panel inset labels ("[model] vs. [reference]") indicate which IDF model's skill is shown and which IDF model is used as reference. The indices refer to the features of the models (c=curvature, m=multiscaling, f=flattening). The skill of the individual features (columns (a)-(c)) are strongly depending on the models that are used, which is why all of these subfigures have to be shown in the manuscript.
We will extend and improve the description of this figure in lines 252 ff. to make it more clear.

**References**

[1] Coles, S. (2001). An introduction to statistical modeling of extreme values. London: Springer-Verlag. ISBN: 1-85233-459-2

[2] Papalexiou, S. M., Koutsoyiannis, D. (2013). Battle of extreme value distributions: A global survey on extreme daily rainfall. Water Resources Research, 49, 187–201.

[3] Van de Vyver, H. (2018). A multiscaling-based intensity–duration–frequency model for extreme precipitation. Hydrol. Process., 32, 1635–1647, https://doi.org/10.1002/hyp.11516,

[4] Shrestha, A., Babel, M. S., Weesakul, S., Vojinovic, Z. (2017). Developing Intensity–Duration–Frequency (IDF) curves under climate change uncertainty: the case of Bangkok, Thailand. Water, 9(2), 145.

---

## Author Response (AR1)

**Author's Response**

Felix S. Fauer, Jana Ulrich, Oscar E. Jurado, Henning W. Rust

October 1, 2021

Dear Xing Yuan,

We hereby submit the new and revised version of our manuscript *Flexible and Consistent Quantile Estimation for Intensity-Duration-Frequency Curves*. We would like to thank you and the reviewers for the many constructive comments which we addressed in the new version. In the following, the comments and our answers are listed point-by-point.

Sincerely,

Felix Fauer (on behalf of all authors)

**1** Reviewer 1 (Rasmus Benestad)**

**Comment 1:** Due to the small samples of the rainfall from clouds which rain gauges with a diameter of centimetres represent, I wonder if it's justifiable to combine data from stations with a distance below 250 m into a single station data set. It's probably OK for the statistical properties, but maybe not so for the raw data - especially on short time intervals (too much random sampling noise?). At least, there ought to be a test of the robustness of the results to this assumption.

Answer: Thank you for this remark. We value transparency in data processing which is why we described carefully how station data was obtained. We agree that the grouping of stations has to be treated carefully. In most cases, data with different temporal resolutions are obtained from different devices. For more transparency, we present the distance between devices of the same station in Tab. 1 (in this document). It shows that the average distance between stations is much lower than 250m. We argue that the grouping should only be relevant for maxima of  $d \ge 24h$  (which can be derived from minutely, hourly or daily data) and on this time scale, the spatial distance should have less impact on the results for most stations.

To verify the robustness of this approach, we investigate the influence of the grouping on the resulting estimated IDF curves. For this purpose, we use for the modeling of maxima with  $d \ge 24h$  (1) the maxima originating from the time series with daily resolution and (2), if available, the maxima originating from the higher resolution time series to estimate the IDF curves. In both cases the time range covers all years in which minutely data is available,

so that the number of used data points is the same. Exemplarily, the two IDF curves for Bevertalsperre and Buchenhofen are shown in Fig. 1. For Buchenhofen, the differences are sufficiently small. For Bever the intensities derived from daily data are smaller, which was expected, since daily sums from daily data come from a fixed time frame (e.g. 00:00 to 23:59) whereas daily sums from minutely data come from a 24-hour window that is shifted minute-by-minute and so, larger sums are captured. When measurements for daily precipitation sums exist from different devices with different temporal resolutions, we decided to keep minutely data in our study because of the flexible time window. For most other stations, IDF curves for both cases do not differ significantly. For few stations, there are differences, especially when only short records are available.

We addressed these remarks in lines 102-106 (in the marked-up manuscript).

A major advantage of this procedure is that the full potential of the duration-dependent GEV model can be used when grouping daily and minutely data. This way, the model can profit from both long daily records and short records with high temporal resolution at the same time and information can be used and transferred efficiently.

Additionally, we will correct a wrong number. The total number of used stations is 115. After the grouping process, 92 stations remain for our analysis (lines 92-93).

**Comment 2:** Perhaps it would make the flow of the text better if there was some connection between the return value (probability) and the quantile? I thought the part on Quantile Skill Index (e.q. 13) wasn't as easy to follow as the preceding sections (how does it link to the preceding discussion on the GEV and the estimation of the parameters?).

Answer: Thank you for pointing out that the part about the Quantile Skill Index was not clear. Sect. 2.4 explains how the model is trained to get the best parameter estimate. Sect. 2.5 describes the verification to evaluate the model's performance and to compare it to other models. The words "quantile" and "return value" are used synonymous. In the manuscript we refer (non-exceedance) "probabilities" *p* to the corresponding "return periods" *T* since they can be easily converted with T = 1/(1-p).

We rephrased the introduction in this subsection (lines 204-206) and linked it to the estimation of GEV parameters.

**Comment 3:** The description of the bootstrapping was a bit difficult to follow - perhaps explain it more carefully or add an illustration?

**Answer:** Thank you for this remark. We improved the description (lines 242-270) and added a reference to Davison & Hinkley (1997) where this procedure is described in detail.

**Comment 4:** Very brief catches (e.g. minutes) of rainfall with rain gauges are expected to be subject to a large degree of sampling uncertainty, aren't they? (depending on the number of rain drops falling onto the cross section representing the measurement). Maybe this also can explain some discrepancies at the extreme short end of the scale?

**Answer:** Thank you for this remark. Minutely measurements might indeed be less accurate when only a small number of rain drops is recorded. However, for events that are identified as annual maxima we expect the rain amount to be large enough that a higher sampling uncertainty compared to larger measurement accumulation sums can be neglected. We added this note in lines 121-123.

**Comment 5:** One reason why annual maxima of different durations do not follow the same scaling process could be that different rain-producing meteorological phenomena have different temporal and spatial scales. If the rainfall can be considered as a 'by-product' of different processes and conditions (e.g. convection, weather fronts, cyclones, and derechoes), then different statistics may perhaps show the true situation? But I'm still struggling to understand what the skill estimates really say.

**Answer:** Thank you for this comment. We agree that maxima for different durations might origin from processes on different spatio-temporal scales. The aim of this new model is to be more flexible and cover extremes from different processes. Especially, it is able to combine the estimation of extremes from both short, mainly convective, processes and processes on longer time scales like derechos. It would be interesting to investigate different statistics for different durations, but then we would need to have a reliable theory that justifies which statistic should be used for which duration. The quantile score (QS) is a proper scoring rule (Gneiting, 2011) and is obtained for each duration individually. Therefore, it can evaluate the model performance, independent of underlying physical processes. Its purpose is to asses how well the model quantiles represent the observed maxima. Skill estimates describe how well the model performs in terms of QS, compared to a reference, especially to the commonly used models IDF or IDFc or IDFm (for model description, see Tab. E1 in the manuscript).

We improved Sect. 2.5 about the verification accordingly to describe the purpose of the skill estimates (lines 203-240).

**Comment 6:** A new and relevant paper DOI: 10.1088/1748-9326/abd4ab suggests a simple formula for expressing IDF curves even for sites with limited data. This formula is based on more 'physical' parameters (wet-day mean precipitation and wet-day frequency), rather than the stronger reliance on the statistical/mathematical theory behind GEV. It would be interesting to compare the results presented here with this formula. It also fits in the comparison of different ways to parametrize IDF curves. At least, it could be included in the introduction and the discussion of different ways of calculating IDFs.

**Answer:** Thank you for suggesting this relevant paper. We included it in the discussion (lines 395-400) where we added a paragraph about non-stationarity, as suggested by another reviewer.

**Comment 7:** Appendices: when describing what calculations and processing was done in this analysis, it's more elegant to use past tense rather than present tense (my subjective

opinion). But mixing past and present tense makes the text inconsistent and a 'clumsy' read. Also check the references therein ('??').

**Answer:** Thank you for these two suggested improvements. We improved both the tenses and the references in the appendix.

**2 Reviewer 2**

**Comment 1:** Line # 29: could be written as Methods **Answer:** Thank you for this suggestion. We changed this sentence (line 29).

**Comment 2:** In lines # 65-70: A few aspects of nonstationarity could be discussed here, discuss briefly the added value of this study to address nonstationarity as compared to the methods discussed in the literature (Cheng and AghaKouchak 2014; Ganguli and Coulibaly 2017).

**Answer:** Thank you for suggesting these relevant papers. We included non-stationarity in the discussion (lines 395-401) because we agree that this is an important concept that should be considered in IDF curves. However, this study is not supposed to add value in non-stationarity, since stationarity is an assumption of our model. We plan to include non-stationarity in future studies.

**Comment 3:** Section 2.1: Lines 90 - 95: This was not very clear - why 8 stations were merged into a single station leaving 92 overall stations out of 100 stations? Which physiographic or hydrologic similarity measures were adopted for regionalization?

**Answer:** Thank you for pointing out that this is not clear. We group the stations because the model can use its full potential, when long daily records are combined with high resolution (minute) records which are often much shorter. Typically, two different measuring frequencies are obtained from different devices and in some cases, those devices are not positioned at the same site. However, in most cases those stations are very near to each other or even at the same position. Therefore, whenever two stations have a distance of less than 250m, those respective stations are merged into one station. The only measure for this procedure is the distance. For more details about the distance of the stations and a test of robustness, see also the answer to comment 1 to Reviewer 1 (Sect. 1).

We described that more clearly in the manuscript (lines 92-106). Also, we corrected a mistake in the manuscript. The total number of stations is 115. The number of grouped stations is 92 (lines 92-93).

**Comment 4:** It has been shown in the literature that the Generalized Maximum likelihood method, in general, does not provide a credible estimate of the shape parameter, yielding an abrupt estimate of shape estimate (Martins and Stedinger 2000). Have your values lie within the credible limits of shape parameter range as shown in the literature, boxplots showing the range of shape parameters for different duration could help to identify this issue

Answer: Thank you for this remark. Since in our model we assume the shape parameter to be constant over duration, we cannot present the range of the shape parameter depending on duration. However, we present the range of the shape parameter at all used stations in Fig. 2 to investigate whether it lies within credible intervals. Almost all parameters lie within  $-0.3 < \xi < 0.6$ . Only for one station (Dabringhausen), an unrealistically large shape parameter was estimated which could be explained by the scarce data availability at this station (15 years).

Martins and Stedinger (2000) report that maximum likelihood estimation (MLE) in small samples can lead to unrealistic shape parameters and that for large sample sizes the RMSE of both methods become similar (Figs. 4 and 5 in their study). We argue that due to the duration-dependent GEV in our study the number of data points available for estimation is multiplied by the number of duration steps and so, sample size should be sufficiently large for using MLE.

**Comment 5:** In skill score index in lines 190-200: what M and R represent, If R denotes empirical distribution, which empirical plotting position formula was used to estimate it? Typically Gringorten's plotting position formula is in use to characterize extremes.

**Answer:** R does not denote the empirical distribution but another IDF model. The quantile skill score (QSS) compares the quantile score ( $QS_M$ ) of a new model M with the quantile score ( $QS_R$ ) of a reference model R. The difference between models and references are only characterized by the features (curvature, multiscaling, flattening) that are used/not used in this model or reference. So, different combinations will be compared, e.g. a model using curvature and multiscaling (IDFcm as model M) vs. a model using only curvature (IDFc as reference model R). Table E1 in the manuscript lists which feature combinations are used as model M and reference R. The quantile score (QS) is a propper scoring function (Gneiting, 2011) for comparing modeled quantiles to all observations. We do not calculate a difference between model quantiles and empirical quantiles. Empirical distributions are only used for visualization ("+" in Figures 5 and 7 in the manuscript).

Thank you for pointing out that this part was not clearly described. We improved Section 2.5 accordingly.

**Comment 6:** Fig. 7: Could you please show the difference in return levels in an inset diagram with vs without flattening? How much is the percentage difference between the two statistics in order to qualify as significant?

**Answer:** We created additional panel plots that show the difference in intensity and also the ratio of both estimates between the two models (Fig. 3 in this document). In comparison with Fig. 5 from the manuscript the differences are expected to overlap with the confidence intervals, suggesting non-significance. However, these confidence intervals are mainly depending on data size and would become smaller with more data available. Nevertheless, the purpose of this visualization was to demonstrate that allowing for flattening of IDF curves in

long durations has an impact on the shape of IDF curves in short durations. These figures only show two selected stations, exemplarily.

In our study, the verification of model performance for all stations was done with the Quantile Skill Index (QSI) where we could show that certain models improve IDF estimation in certain duration-regimes, depending on the use-case. Here, a QSI

Figure 1: Comparison of IDF curves for two different ways of obtaining maxima for  $d \ge 24h$ . keep min: all maxima are derived from minutely data. keep day: maxima  $d \ge 24h$  are derived from daily data.

Figure 2: A histogram, showing the values of shape parameters from all stations. The shape parameter is constant across durations. The median is shown with a red line.

| Index | Number | Station name                     | Distance (m) |
|-------|--------|----------------------------------|--------------|
| 1     | 3      | Breckerfeld-Wengeberg            | 90           |
| 2     | 16     | Hückeswagen (Bevertalsperre)     | 101          |
| 3     | 18     | Köln-Stammheim                   | 49           |
| 4     | 30     | Remscheid-Lennep                 | 34           |
| 5     | 32     | Solingen-Hohenscheid             | 22           |
| 6     | 35     | Wipperfürth-Gardeweg             | 52           |
| 7     | 37     | Wuppertal-Buchenhofen            | 217          |
| 8     | 50     | Bochum                           | 0            |
| 9     | 51     | Dormagen-Zons                    | 0            |
| 10    | 53     | Köln-Bonn                        | 0            |
| 11    | 54     | Essen-Bredeney                   | 0            |
| 12    | 64     | Reichshof-Eckenhagen             | 0            |
| 13    | 65     | Neunkirchen-Seelscheid-Krawinkel | 0            |
| 14    | 66     | Lüdenscheid                      | 0            |
| 15    | 67     | Meinerzhagen-Redlendorf          | 0            |
| 16    | 68     | Overath-Böke                     | 0            |
| 17    | 69     | Gevelsberg-Oberbröking           | 0            |
| 18    | 72     | Leverkusen                       | 4            |
| 19    | 74     | Neumühle                         | 37           |
| 20    | 75     | Schwelm                          | 24           |

Table 1: Distance between devices of grouped stations. Two bold stations are chosen in the manuscript for detailed IDF curves.

---

## Author Response (AR2)

**Author's Response**

Felix S. Fauer, Jana Ulrich, Oscar E. Jurado, Henning W. Rust

November 15, 2021

Dear Xing Yuan,

We hereby submit the new and revised version of our manuscript *Flexible and Consistent Quantile Estimation for Intensity-Duration-Frequency Curves*. We would like to thank you and the reviewers for your constructive comments which we addressed in the new version. In the following, the comments and our answers are listed point-by-point.

Sincerely,
Felix Fauer (on behalf of all authors)

**Comment 1:** Please clarify the the limitations of the study (e.g. scarce data availability precludes the reasonable estimation of the shape parameter).
**Answer:** Thank you for this remark. In lines 371-375, we added a short paragraph about the limitations of the shape parameter estimation and a possible solution on how scarce data availability can be dealt with in extreme value analysis.

**Comment 2:** Provide a broader implications of the study.
**Answer:** Thank you for pointing out that broader implications of the study were not clear. These changes were made: In lines 49-51 we shortly describe the advantage of a model extension. In line 381 we added a short remark about broader implications. In lines 388-390 we explain why improving efficient estimation is important and how these insights can be used.